# The Effect of a Personalized Approach to Patient Education on Heart Failure Self-Management

**DOI:** 10.3390/jpm8040039

**Published:** 2018-11-27

**Authors:** Muhammad W. Athar, Janet D. Record, Carol Martire, David B. Hellmann, Roy C. Ziegelstein

**Affiliations:** 1Division of Cardiovascular Health and Diseases, University of Cincinnati College of Medicine, 231 Albert Sabin Way, ML 0542, Cincinnati, OH 45267-0542, USA; mw.athar@gmail.com; 2Department of Medicine, Johns Hopkins Bayview Medical Center, Johns Hopkins University School of Medicine, 5200 Eastern Avenue, MFL building East Tower, 2nd floor, Baltimore, MD 21224, USA; cmartire@jhmi.edu (C.M.); hellmann@jhmi.edu (D.B.H.); rziegel2@jhmi.edu (R.C.Z.)

**Keywords:** heart failure, adherence, portable ultrasound, patient education, hospital readmission

## Abstract

Personalized tools relevant to an individual patient’s unique characteristics may be an important component of personalized health care. We randomized 97 patients hospitalized with acute decompensated heart failure to receive a printout of an ultrasound image of their inferior vena cava (IVC) with an explanation of how the image is related to their fluid status (*n* = 50) or to receive no image and only generic heart failure information (*n* = 47). Adherence to medications, low-sodium diet, and daily weight measurement at baseline and 30 days after discharge were assessed using the Medical Outcomes Study Specific Adherence Scale, modified to a three-item version for heart failure (HF), (MOSSAS-3HF, maximum score = 15, indicating adherence all of the time). The baseline MOSSAS-3HF scores (mean ± standard deviation (SD)) were similar for intervention and control groups (7.4 ± 3.4 vs. 6.4 ± 3.7, *p* = 0.91). The MOSSAS-3HF scores improved for both groups but were not different at 30 days (11.8 ± 2.8 vs. 11.7 ± 3.0, *p* = 0.90). Survival without readmission or emergency department (ED) visit at 30 days was similar (82.6% vs. 84.1%, *p* = 0.85). A personalized HF tool did not affect rates of self-reported HF treatment adherence or survival without readmission or ED visit.

## 1. Introduction

Episodes of acute decompensated heart failure (ADHF) account for the largest proportion of admissions and 30-day readmissions to US hospitals [1]. Several patient-related factors, especially nonadherence to heart failure (HF) medications and a low-sodium diet, have been shown to play an important role in preventable HF readmissions [2,3]. Personalized patient education, rather than generic advice, may be perceived as more relevant and thus may be more effective in motivating patients to change behaviors that influence health [4]. Personalizing patient education by sharing a patient’s own medical images that elucidate an active disease process, with visual evidence of physical harm which is directly related to a certain behavior, may enhance patients’ motivation to change the risky behavior [5,6]. For example, sharing with patients ultrasound images of their own large arteries with atherosclerotic plaques has been shown to improve smoking cessation rates [7]. Another group found that more severe atherosclerosis on diagnostic imaging correlated with enhanced short-term commitment to stop smoking [8]. Among patients with HF, picture-based educational materials, as part of a multimodal intervention, were associated with lower rates of death and rehospitalization, and higher rates of adherence to daily weight monitoring [9]. We hypothesized that personalized visual images, using information of unique relevance to each patient with HF, may have even greater potential for motivating patients to pursue healthy behavior changes and reduce nonadherence to medications and diet. For example, sharing with patients images of their own coronary artery calcium scans obtained by cardiac computed tomography, along with an explanation of the images and education on behavioral risk factor modification, has been associated with greater levels of patient adherence to medications and dietary modifications among patients with coronary artery disease [10,11].

A major cause of acute decompensation in patients with HF is volume overload, often resulting from a failure to adhere properly to HF medications or to a low-sodium diet. A patient’s volume status can be accurately assessed with ultrasound images of the inferior vena cava (IVC). In particular, a dilated IVC with poor collapsibility during inspiration correlates with an elevated central venous pressure. Among patients hospitalized for ADHF, a persistently dilated IVC with poor collapsibility prior to hospital discharge has been associated with higher readmission rates for ADHF [12]. We aimed to personalize an approach to educating patients with ADHF by providing patients with a printout of an ultrasound image of their IVC along with scripted education describing how that image, and their volume status, were influenced by adherence to HF medications and a low-sodium diet. We hypothesized that this personalized approach would improve HF self-management and reduce ADHF readmissions.

## 2. Methods

### 2.1. Design and Participants

This was a prospective, randomized controlled trial on a general inpatient medicine service of a 447-bed, urban academic medical center. Adult, non-pregnant patients admitted between March and August 2015 and receiving intravenous diuretics for the treatment of ADHF were enrolled after 36–60 h in the hospital. This time window was chosen to allow for verification of the primary diagnosis and treatment indication. Patients who were unable or unwilling to provide informed consent, unable to read or speak English, or without a working phone number were excluded. The exclusion of patients unable to provide consent due to cognitive limitations ensured that patients in the study had the cognitive ability to process educational information regarding heart failure. Patients with recent abdominal surgery or with severe abdominal pain or tenderness were excluded to avoid patient discomfort due to pressure from the ultrasound probe. Patients who were readmitted more than 30 days after discharge from the index hospitalization were eligible to participate again.

### 2.2. Study Procedures

A randomization sequence was generated in sets of six for a total of 100 participants using Research Randomizer (https://www.randomizer.org/). Sealed envelopes were used to conceal the study group assignment, and this step was completed before the enrollment of the first patient. The study ultrasonographer had no role in the randomization or initial screening of electronic medical records to identify eligible patients.

Each patient underwent measurement of baseline adherence using the Medical Outcomes Study Specific Adherence Scale [13] modified to a three-item version focused on HF (MOSSAS-3HF). The MOSSAS-3HF asks patients to rate how often in the past four weeks they adhered to their medication regimen, low-sodium diet, and daily weight measurement. The answer choices ranged from 0 (none of the time) to 5 (all of the time), with a maximum total score of 15. All patients underwent a limited bedside ultrasound examination of the IVC (measuring maximum IVC diameter (IVC_max_) and IVC collapsibility index (IVCCI)) by an experienced, board-certified ultrasonographer using a MicroMaxx portable ultrasound machine (Sonosite^®^ Inc., Bothel, WA, USA) with a P17-5MHz ultrasound probe at subxiphoid placement. Measurements taken were at 1.0 to 2.0 cm from the IVC/right atrium juncture, taking care to avoid any hepatic veins emptying into the IVC at the approximation of the IVC/right atrium interface. Patients were asked to “sniff” to measure the IVC collapsibility with inspiration.

After the baseline MOSSAS-3HF responses were obtained, just prior to conducting each limited bedside portable ultrasound examination, the patient’s randomization status into the attention control group (CG) or the educational intervention group (IG) was revealed to the study ultrasonographer/research assistant (C.M.) by opening a sealed envelope, as above. Only IG patients were shown their IVC images by the ultrasonographer, who also provided them with real-time scripted educational information. The ultrasonographer/research assistant is a board-certified sonographer and has completed training in patient education and research methods. All study team members agreed upon the verbal patient education script. One study team member (M.W.A.) reviewed the delivery of the script with the ultrasonographer/research assistant (C.M.). The patient education emphasized the relationships among a distended IVC, episodes of ADHF, and adherence to HF medications and a low-sodium diet. While a consistent approach to verbal patient education was maintained, with attention to the use of language that would be clear to individuals with low health literacy levels as well as higher levels, the patient education was personalized in several ways. For example, the ultrasonographer/research assistant asked each patient about their favorite foods and other habits, personalizing the dietary advice based on what was learned by the responses to those questions. Patient education was also tailored to the degree of the distention of the patient’s IVC. For patients with a distended IVC, the connection with excess dietary sodium, causing fluid retention and the episode of ADHF, was straightforward. For patients with IVC diameters in a normal or near-normal range, which may be seen soon after administration of diuretics for ADHF, education was tailored to reflect that these visual findings suggested a good response to medical treatment for ADHF, and that earlier in their hospital stay their images likely would have been different. In order to ensure that patients understood the content of the teaching, the ultrasonographer/research assistant asked each patient to restate their understanding of what was discussed, and to provide their understanding of next steps in self-care.

The IG also received a printed and laminated Patient Education Tool (Appendix A) with sample IVC images, space for a printout of the patient’s own IVC image, and education about HF self-care. The sample images on the Patient Education Tool included both a dilated IVC (consistent with ADHF) and normal IVC (which may be seen after effective treatment for the episode of ADHF). Text in the Patient Education Tool explained links between an abnormal, dilated IVC as that seen with ADHF and behaviors that led to ADHF episodes. Actionable steps to prevent ADHF were listed, such as specific strategies to reduce sodium intake and reminders of the importance of medication adherence and weight monitoring. The sample image of the dilated IVC provided patients with visual evidence of the disease process of ADHF. This visual evidence was meant to be a source of motivation for the key behavior changes listed on the Tool. Language in the Patient Education Tool was written at approximately a fourth-grade level complexity in order to be useful to patients across a wide range of literacy levels.

The control group patients also underwent ultrasound imaging; however, the portable ultrasound screen was positioned so that they could not view their own IVC images. The study ultrasonographer followed a script for CG patients that excluded information about IVC imaging. The attending physicians and other clinicians caring for patients in both groups were aware of patients’ participation in the study, but were blinded to the results of the portable ultrasound exam. The clinicians managed and discharged patients based on information obtained as part of routine clinical care, without any input from the study team. The clinicians were notified of findings of potential clinical importance when applicable. The patients in both IG and CG received a generic HF education booklet (not related to IVC imaging), as is done for all patients with ADHF as part of the standard of care at the study institution. The study protocol was approved by the Johns Hopkins University Institutional Review Board (protocol identification number IRB00034300).

### 2.3. Outcome Measures

The primary outcome measure was HF regimen adherence 30 days after hospital discharge, as assessed by the MOSSAS-3HF, administered by telephone. The study team research assistant (C.M.) performed the follow-up telephone calls, following a script to minimize bias. Review of the electronic medical records provided one source of information to determine survival without readmission or urgent care visit within 30 days. To account for any readmissions or urgent care visits at other health care institutions, we used direct questioning with a standardized script during the follow-up telephone call to patients at 30 days. The sample size was determined using data from the administration of the MOSSAS-3HF to a pilot sample of patients hospitalized with ADHF to inform a power calculation. Based on these preliminary data, the power to detect a one-point change in the MOSSAS-3HF scores was estimated at 80% with 5% alpha error if the sample size was 100 patients (50 patients in CG and 50 patients in IG).

## 3. Results

### 3.1. Enrollment and Baseline Data

A total of 206 patients admitted with ADHF were screened; 79 did not meet all inclusion criteria, 10 declined participation, nine were discharged before obtaining informed consent, eight were deemed not appropriate by their treating physician (due to language barrier, psychiatric illness, or a social barrier such as homelessness), and three did not have adequate IVC images. A total of 97 eligible patients signed informed consent and were randomized to either CG (*n* = 47) or IG (*n* = 50, Figure 1). Fifty-six patients (59.6%) were women. Age, left ventricular ejection fraction, and comorbidities were equally distributed among both groups, except that diabetes mellitus was more prevalent in CG (61.7% vs. 36.0%, *p* = 0.01) (Table 1). The control group and IG had similar IVC_max_ values (mean ± standard deviation (SD)) (2.43 cm ± 0.47 vs. 2.31 cm ± 0.46, *p* = 0.21) and IVCCI values (34.4% ± 20.1 vs. 34.5% ± 28.9, *p* = 0.98) at the time of enrollment in the study, indicating a similar volume status in both groups at study entry. Notably, an IVC diameter of 1.5–2.5 cm with less than 50% collapsibility corresponds to a central venous pressure (CVP) of 11–15 cm, which is elevated [14].

The baseline MOSSAS-3HF scores between groups were not different (CG 6.4 ± 3.7 vs. IG 7.4 ± 3.4, *p* = 0.91). The telephone follow-up at 30 days was completed in 44 patients in CG (one died; two could not be contacted by telephone, one of whom was known to be alive based on the electronic medical record) and in 46 patients in IG (one died; three could not be contacted by telephone but were known to be alive based on review of the medical record).

### 3.2. Outcomes

Patients’ self-reported levels of adherence to HF treatment as measured by MOSSAS-3HF scores were not different between the CG and IG groups 30 days after discharge (11.7 ± 3.0 vs. 11.8 ± 2.8, *p* = 0.90) (Table 2). There was no difference in 30-day survival without readmission or ED visit between CG and IG (84.1% vs. 82.6%, *p* = 0.85). The MOSSAS-3HF scores increased from baseline to 30 days in both groups (IG, 7.4 to 11.8; CG, 6.4 to 11.7 (*p* < 0.001 for both)). Among patients who had no 30-day readmissions or urgent care visits, the mean follow-up MOSSAS-3HF scores were higher than those of patients who required readmission or urgent care visits within 30 days (12.1 ± 2.6 vs. 9.8 ± 3.8, *p* = 0.04); baseline MOSSAS-3HF and *IVC*_max_ values were not different between these groups.

## 4. Discussion

Providing a printout of an individual’s ultrasound images of their IVC to patients admitted with ADHF, along with a personalized, scripted explanation of the relationship of those images to volume status, did not improve self-reported HF self-management or affect 30-day survival without readmission or urgent care visit. There has been some prior published experience with personalized educational tools in HF patients. Most notable is a study that showed that an intervention that included picture-based educational materials, a digital scale, and telephone calls intended to reinforce adherence as well as reduce hospitalization or death over a 12-month period [9]. That intervention differed from ours in many important respects, however. It was delivered to outpatients rather than to hospitalized patients, and it included nearly weekly follow-up telephone calls for 8 weeks, followed by monthly calls for several months after that. Patients were instructed on diuretic self-adjustment, and those who experienced worsening symptoms were scheduled acute visits with their physician. Other “high touch” personalized care management strategies designed specifically for HF patients [15] or for individuals with chronic illness [16] (some of whom had HF) have also shown promise. However, other interventions in HF patients have failed to demonstrate improved outcomes, including individualized health coaching telephone calls [17] and adherence telemonitoring, in which a licensed clinical social worker contacted and provided education to nonadherent HF patients after hospital discharge [18]. While these interventions used personalized educational tools, to our knowledge our intervention is the first in HF patients to provide actual patient images to study participants. Notably, a prior study showed individuals computed tomography scans of their heart that demonstrated coronary artery calcium as evidence of atherosclerosis, to motivate them to initiate aspirin therapy and dietary changes, and to increase their level of exercise [11].

Personalized medicine is an approach that accounts for each patient’s health beliefs, understanding of their health conditions, trust in recommendations from health care providers, and life circumstances that allow for adherence to recommendations [19]. We hypothesized that a personalized medicine approach would have an important impact on HF outcomes since these outcomes are known to be influenced by behaviors that are directly controlled by patients—namely, adherence to medical treatment and a low-sodium diet. However, an effect of the personalized educational tool used in this study on HF self-management was not demonstrated in this study. One reason for this is that self-management improved in both groups of patients. Although the attention control arm in this study was designed to control only for specific components of the intervention that we assumed would not influence HF self-management (e.g., we controlled for the time spent with the patient by the sonographer and the actual performance of the ultrasound examination), it is possible that even this attention and human connection was perceived as personalized care and was sufficient to improve patient adherence. In essence, it is possible that a personalized approach was evinced in both the intervention and control groups, unintentionally making it unlikely to demonstrate a specific contribution of the shared IVC image. It is also possible that the standard, written HF education material given to all patients with ADHF at the study hospital may also have enhanced adherence for all patients. It is also possible that the degree of personalization of our intervention could have been greater if we had had the ability to capture ultrasound images of each patient’s IVC at the time of the initial presentation, prior to the initiation of treatment with diuretics. If this timing had been possible, more patients may have had a greater degree of IVC distention, and the personalization of the messages in the Patient Education Tool may have been more impactful in motivating behavior change in the IG group. In addition, we aimed to deliver the scripted educational information in a consistent manner, and the verbal and written language were designed to accommodate patients with lower literacy levels. Because we aimed for consistency, we did not increase the complexity of our patient education language for those patients with higher literacy levels. This factor may have limited the personalization and impact of our intervention. However, the potential value of an improvement in adherence due to a personalized medicine approach is suggested by the finding that follow-up MOSSAS-3HF scores were higher for patients surviving without hospital readmission or requiring urgent care visits within 30 days.

Despite randomization procedures, the prevalence of diabetes was higher in the CG group. Patients with diabetes who seek regular medical care generally receive self-care education on dietary modifications and, in many cases, on self-monitoring via regular glucose monitoring. Such patients may be more accustomed to, and possibly more skilled at, implementing recommendations on self-management and monitoring. Thus, it is possible that the increased prevalence of diabetes in CG diminished our ability to detect an impact of our personalized HF educational intervention.

The limitations of this study included a small sample size, single institution setting, and reliance on patients’ self-report of adherence and acute care visits. Concrete behavioral measures of patient adherence to medications, low-sodium diet, and daily weight measurement were not possible. Additionally, the performance of the portable ultrasound for patients in the control arm was almost certainly not a neutral attention control design, which may have hindered our ability to detect a difference between groups.

## 5. Conclusions

Personalized HF education in which patients were shown their own IVC images did not improve HF self-management or affect survival without readmission or urgent care visit 30 days after discharge. Higher 30-day MOSSAS-3HF scores were observed in patients with fewer hospital readmissions or urgent care visits, suggesting that the MOSSAS-3HF may be a useful measure of HF self-management.

## Figures and Tables

**Figure 1 jpm-08-00039-f001:**
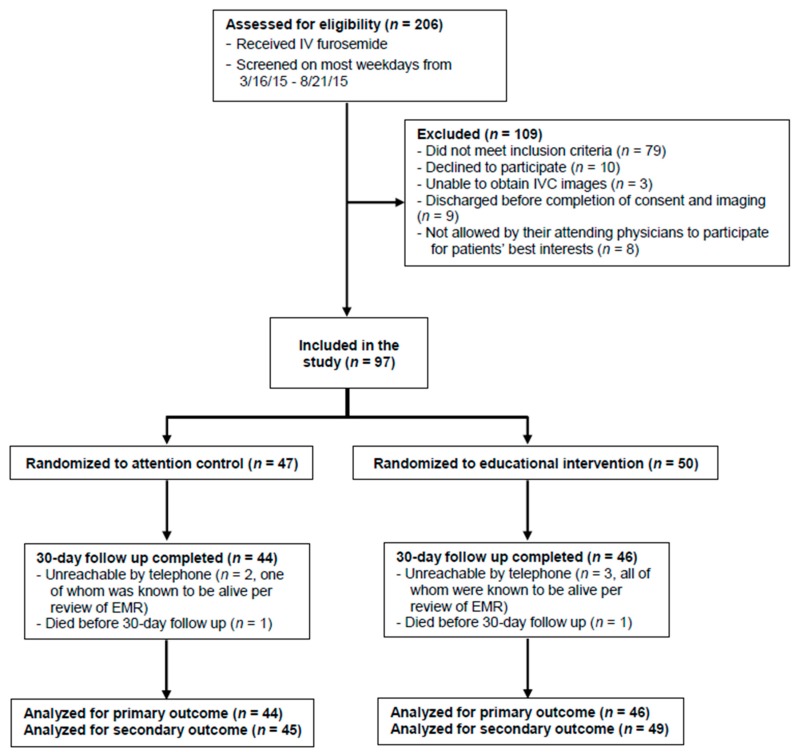
Study flowchart. Participant flow is shown for this randomized trial of a personalized educational intervention using ultrasound images of patients’ own inferior vena cava dimensions for hospitalized patients with acute decompensated heart failure. Abbreviations: IV, intravenous; IVC, inferior vena cava; EMR, electronic medical record; ED, emergency department. Primary outcome was degree of heart failure regimen adherence 30 days after discharge. Secondary outcome was survival without readmission or emergency department visit within 30 days of discharge.

**Table 1 jpm-08-00039-t001:** Baseline characteristics.

Characteristic	Control Group (CG) (*n* = 47)	Intervention Group (IG) (*n* = 50)	*p* Value
Female, *n* (%)	28 (59.6)	28 (56.0)	0.72
Age, years, mean (SD)	72.4 (11.7)	70.3 (13.7)	0.43
**Comorbidities**			
Hypertension, *n* (%)	41 (87.2)	47 (94.0)	0.31
Diabetes mellitus, *n* (%)	29 (61.7)	18 (36.0)	0.01
Coronary artery disease, *n* (%)	22 (46.8)	28 (56.0)	0.37
Atrial fibrillation, *n* (%)	19 (40.4)	20 (40.0)	0.97
Chronic kidney disease (CrCl < 60 mL/min), *n* (%)	22 (46.8)	19 (38.0)	0.38
Left ventricular ejection fraction, ^a^ mean % (SD)	46.5 (17.5)	45.8 (16.1)	0.84
**IVC Measurements**			
IVC_max_ (cm), mean (SD)	2.43 (0.47)	2.31 (0.46)	0.21
IVC_min_ (cm), mean (SD)	1.66 (0.75)	1.56 (0.70)	0.52
IVCCI (%), mean (SD)	34.4 (20.1)	34.5 (28.9)	0.98
**Qualitative IVC Impression**			
Dilated and poorly collapsible (IVC > 2.5 cm diameter with poor phasicity; correlates with CVP > 15 mm Hg), *n* (%)	22 (46.8)	21 (42.0)	0.88
Normal diameter with <50% collapse (IVC 1.5–2.5 cm with poor collapsibility; correlates with CVP 10–15 mm Hg), *n* (%)	7 (14.9)	9 (18.0)	
Normal diameter with >50% collapse (IVC 1.5–2.5 cm with normal collapsibility; correlates with CVP 5–10 mm Hg), *n* (%)	15 (31.9)	18 (36.0)	
Small and very collapsible (IVC < 1.5 cm with ≥50% collapse); correlates with CVP 0–5 mm Hg), *n* (%)	3 (6.4)	2 (4.0)	

Abbreviations: SD, standard deviation; ADHF, acute decompensated heart failure; CrCl, creatinine clearance; IVC, inferior vena cava; IVC_max_, inferior vena cava maximum diameter; IVC_min_, inferior vena cava minimum diameter; IVCCI, inferior vena cava collapsibility index; CVP, central venous pressure. ^a^ Last known ejection fraction available in the electronic medical record. For one patient in the control group and two patients in the intervention group, no ejection fraction data were available.

**Table 2 jpm-08-00039-t002:** Results 30 days ^a^ after discharge.

	**Control Group (CG)** **(*n* = 44)**		**Intervention Group (IG)** **(*n* = 46)**		***p* Value for 30-Day Comparison**
	**Baseline**	**30 Days**	**Baseline**	**30 Days**	
**Total MOSSAS-3HF**^b^, range 0–15, mean (SD)	6.4 (3.7)	11.7 (3.0)	7.4 (3.4)	11.8 (2.8)	0.90
**Individual MOSSAS-3HF items**					
Item 1: Took medication as prescribed (on time without skipping dose) (range 0–5), mean (SD)	3.2 (1.6)	4.7 (0.5)	3.2 (1.8)	4.4 (1.3)	0.15
	**Control Group (CG)** **(*n* = 44)**		**Intervention Group (IG)** **(*n* = 46)**		***p* Value for 30-Day Comparison**
	**Baseline**	**30 Days**	**Baseline**	**30 Days**	
Item 2: Followed a low-salt diet (range 0–5), mean (SD)	1.9 (1.2)	3.8 (1.0)	2.4 (1.2)	4.0 (1.0)	0.16
Item 3: Weighed yourself every day (range 0–5), mean (SD)	1.3 (1.8)	3.3 (2.0)	1.8 (1.7)	3.4 (1.7)	0.85
Survival without hospital readmission within 30 days, *n* (%)		37 (84.1)		39 (84.8)	0.93
Survival without readmission or ED visit within 30 days, *n* (%)		37 (84.1)		38 (82.6)	0.85
Time (days) from discharge to readmission or ED visit, mean (SD)		9.7 (6.1)		15.4 (9.3)	0.20

Abbreviations: SD, standard deviation; ED, emergency department. ^a^ Follow-up telephone calls were completed in 44 patients in CG and 46 patients in IG (see text). ^b^ MOSSAS-3HF, Medical Outcomes Study Specific Adherence Score—three items for heart failure. Patients are asked how often, in the last 4 weeks, they have completed each behavior. For each item, a patient’s self-reported score of 0 indicates “none of the time”, 1 “a little of the time”, 2 “some of the time”, 3 “a good bit of the time”, 4 “most of the time”, and 5 “all of the time”.

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
