# Peer review of "The Effect of a Personalized Approach to Patient Education on Heart Failure Self-Management"

_jpm, 2018, doi:10.3390/jpm8040039_

Round 1

Reviewer 1 Report

The study examines the impact of personalized advice based on ultrasound images on HF outcomes and behaviors.

Overall the article is well-written and flows well. However limitations in the personalized approach as well as lack of theoretical justification for the intervention should be discussed in the article. My other comments are as below:

Using a theoretical basis to justify the linkage of visual images to behavior change would be more powerful.

Was the provided scripted education tailored or personalized to literacy levels? How was patients' understanding of the information provided to them assessed?

How was patients' cognitive ability to process the information determined?

Did all the IG participants in this study have similar IVC measures?

If not, how was the scripted information tailored to adapt to the different IVC measures?

Who trained the ultrasonographer to provide the scripted education?

How was fidelity to the personalized dietary advice ensured?

How are the sample IVC images provided to IG participants actionable for the patients to make behavior changes?

Was readmission or urgent care visit possible at a different healthcare institution other than the one in which the EMR was available?

How many were newly diagnosed HF patients? How might have a new diagnosis of HF made an impact on the update of behaviors post-discharge as well as other results?

Was a power analysis conducted to determine the sample size?

Discussion could include how much the disproportional prevalence of diabetes might have made an impact on self-care behaviors in the CG

Please address the limitations of the current personalized approach used in the study. “However, an effect of the personalized 190 educational tool used in this study on HF self-management was not demonstrated in this study.”

This may be because the intervention was not personalized enough - especially to the literacy and cognitive ability of the participants? 

Author Response

Reviewer comments

Authors’ responses

1

The study examines the impact of personalized advice based on   ultrasound images on HF outcomes and behaviors.

Overall the article is well-written and flows well. However   limitations in the personalized approach as well as lack of theoretical justification   for the intervention should be discussed in the article. My other comments   are as below.

Thank you for this comment on the flow and writing style.

We will address the questions about limitations in the personalized   approach and need for additional theoretical justification for the   intervention below, in item numbers 13 and 2, respectively.

2

Using a theoretical basis to justify the linkage of visual images to   behavior change would be more powerful.

Additional comments and references to prior work have been added to   the introduction, lines 32-43.

3

Was the provided scripted education tailored or personalized to literacy   levels?  

How was patients' understanding of the information provided to them   assessed?

A consistent approach to verbal patient education was maintained,   with attention to using language that would be clear to individuals with low   health literacy levels and would be accessible to all participants; this   detail was added in lines 106-107.

The language on the printed patient education tool was at a   Flesch-Kincaid grade level of 3.7; clarification in the manuscript was added   to indicate reading level of the printed educational material for patients;   see lines 130-131.

In addition, as per lines 69-70, inclusion criteria required that   patients in the study spoke English and were able to read.  

In order to ensure patients understood the content of the teaching,   the ultrasonographer / research assistant asked each patient to restate their   understanding of what was discussed, and to provide their understanding of   next steps in self-care; see lines 116-119.

4

How was patients' cognitive ability to process the information   determined?

Our inclusion and exclusion criteria served to select only patients   whose cognitive states would allow them process educational information on   heart failure.  We excluded patients   who were unable to consent to participate in the study. Clarification is provided   in lines 71-73.  In addition, please   also see the last portion of our response to item number 3 (assessing   patient’s understanding of information).   

5

Did all the IG participants in this study have similar IVC   measures?  If not, how was the scripted   information tailored to adapt to the different IVC measures?

Since IVC measurements varied among IG (and CG) patients, as shown in   Table 1, section on Qualitative IVC impression, scripted educational   information for IG patients was adapted to indicate the relationship between   treatment of ADHF with diuretic medication and IVC size in the near-normal   range; clarification has been provided in lines 111-116. 

The Patient Educational Tool contained 2 sample images, one with IVC   dilatation (ADHF) and normal IVC (may be seen after medical treatment of   ADHF); this explanation was added in line 123.

6

Who trained the ultrasonographer to provide the scripted education?

The study ultrasonographer had training as a research assistant,   including patient education and communication skills.  All study team members agreed upon the   verbal patient education script. One study team member (MWA) reviewed   delivery of the script with the ultrasonographer / research assistant (CM).   These details were added (see lines 101-104).

7

How was fidelity to the personalized dietary advice ensured?

We were only able to measure patient self-report of adherence to   dietary advice (and to medications and daily weight).  We added clarity in the Limitations section   that we were only able to use self-report for these outcomes, rather than   concrete behavioral outcome measurements. (Lines 274-276)

8

How are the sample IVC images provided to IG participants actionable   for the patients to make behavior changes?

Sample IVC images were provided on the Patient Education Tool that   included specific actionable steps to prevent episodes of ADHF, including   specific strategies to reduce sodium intake, medication adherence, and weight   monitoring. IVC images were meant to be a source of motivation and a reminder   about the importance of HF self-management behaviors.  See lines 122-128.

9

Was readmission or urgent care visit possible at a different   healthcare institution other than the one in which the EMR was available?

Yes, through direct questioning of patients during the 30-day   follow-up telephone call.  This study   procedure is now described in more detail; see lines 146-150.

10

How many were newly diagnosed HF patients? How might have a new   diagnosis of HF made an impact on the update of behaviors post-discharge as   well as other results?

Our patient population included both newly diagnosed and previously   known heart failure patients. We did not categorize patients in terms of new   versus previously known diagnosis. This is an interesting question and we   would consider this aspect in our future studies.

11

Was a power analysis conducted to determine the sample size?

We administered the MOSSAS-3HF to a pilot sample of 5 inpatients   admitted for ADHF.  Based on these   total MOSSAS-3HF scores, and the standard deviation measures we found using   those data, enrolling 100 total patients (50 patients in IG and 50 patients   in CG) would provide power to detect a difference of 0.95 points on the   MOSSAS-3HF.  This information has been   added in lines 143 – 147.

12

Discussion could include how much the disproportional prevalence of   diabetes might have made an impact on self-care behaviors in the CG

Thank you for this suggestion.    Discussion on this factor has been added; see lines 252 – 257.   

13

Please address the limitations of the current personalized approach   used in the study. “However, an effect of the personalized 190 educational   tool used in this study on HF self-management was not demonstrated in this   study.”

Thank you for this suggestion. We have now addressed additional   potential limitations in lines 247 – 252.

14

This may be because the intervention was not personalized enough -   especially to the literacy and cognitive ability of the participants?

We believe that we screened patients to include only those with the   cognitive ability to understand the education involved in our study (see item   4).  We also only included patients who   were able to read English. We believe that we adjusted our verbal and written   language to a sufficiently low grade level to accommodate patients with   limited literacy and limited health literacy (see item 3). 

Because we aimed to deliver the scripted information in a consistent   manner (item 3), we did not increase the complexity of our patient education   intervention for those patient for whom this may have been appropriate. This   factor may have limited the personalization and impact of our intervention.   This information has been added in lines 252 – 256. 

Reviewer 2 Report

Thank you for this succinct and easy to read manuscript that follows logical progression. To enable reproducibility of the study, there are a few items in the method that need to be clarified:

2.1 Please include information about ethics approval gained, how the participants were randomised and how the potential bias of the ultrasonographer were managed.

Who undertook the follow-up telephone calls (2.3)? and due to the nature of the survey, how was the potential for respondent bias minimised (L157-161)?

3.1 What type of reasons did the treating physician give for 'feeling' patients were not appropriate.  This behaviour could contribute to bias in the study and needs to be clarified.

Figure Study flow chart - Furosemide requires a capital letter F.

There is a random change in font size within the manuscript, I assume will be remediated in the production process?

Author Response

Reviewer comments

Authors’ responses

1

Thank you for this succinct and easy to read manuscript that follows   logical progression. To enable reproducibility of the study, there are a few   items in the method that need to be clarified

Thank you for your comments and suggestions. Please see our responses   and modifications to the manuscript below.

2

Please include information about ethics approval gained, how the   participants were randomised and how the potential bias of the ultrasonographer   were managed.

Johns Hopkins University Institutional Review Board reviewed the   study protocol and approved study methods after finding them to be in   compliance with all applicable statutes related to ethical research involving   human subjects. (Lines 140 - 141)

Randomization   sequence was generated in sets of 6 for a total of one hundred participants   by using Research Randomizer (https://www.randomizer.org/). Sealed envelopes   were used to conceal study group assignment, and this step was completed   before enrollment of the first patient. Envelopes were only unsealed after   informed consent and baseline MOSSAS-3HF survey responses were obtained.   (Lines 78-80)

The ultrasonographer had no role in randomization or initial   screening of electronic medical records to identify eligible patients. (Lines   80-82)

The ultrasonographer was blinded to study group assignment until the   envelopes were unsealed after informed consent and baseline MOSSAS-3HF   responses were obtained.  (lines   95-98).

3

Who undertook the follow-up telephone calls (2.3)? and due to the   nature of the survey, how was the potential for respondent bias minimised   (L157-161)?

Our team’s research assistant (CM) performed the follow-up telephone   calls, following a script to minimize bias. (Lines 144-145)

4

What type of reasons did the treating physician give for 'feeling'   patients were not appropriate.  This   behaviour could contribute to bias in the study and needs to be clarified.

Attending physicians in our department were made aware of exclusion   criteria of the study prior to enrollment of the first patient. Attending   physicians deemed 8 patients not to be appropriate for research study due to   language barrier not identified on initial screening of electronic medical   record by study team, acute psychiatric illness or social barriers to adhere   to study protocol like homelessness (lines 159-160)

5

Figure Study flow chart - Furosemide requires a capital letter F.

In the United States, standard practice is not to capitalize generic   medication names, but we will leave this to the discretion of the editors.

6

There is a random change in font size within the manuscript, I assume   will be remediated in the production process?

Thank you. We have made this change to ensure uniform formatting.